# Easily Applicable Predictive Score for Differential Diagnosis of Prefibrotic Primary Myelofibrosis from Essential Thrombocythemia

**DOI:** 10.3390/cancers15164180

**Published:** 2023-08-20

**Authors:** Danijela Lekovic, Andrija Bogdanovic, Marta Sobas, Isidora Arsenovic, Mihailo Smiljanic, Jelena Ivanovic, Jelena Bodrozic, Vladan Cokic, Natasa Milic

**Affiliations:** 1Clinic of Hematology, University Clinical Center Serbia, 11000 Belgrade, Serbia or andrija.bogdanovic@med.bg.ac.rs (A.B.); isidoraarsenovic@yahoo.com (I.A.); mikesmiljanic79@gmail.com (M.S.); jivanovic09@gmail.com (J.I.); bodrozic.jelena@gmail.com (J.B.); 2Faculty of Medicine, University of Belgrade, 11000 Belgrade, Serbia; 3Department of Hematology, Blood Neoplasms and Bone Marrow Transplantation, Wroclaw Medical University, 50-367 Wroclaw, Poland; marta.sobas@gmail.com; 4Institute for Medical Research, University of Belgrade, 11000 Belgrade, Serbia; vl@imi.bg.ac.rs; 5Institute of Medical Statistics & Informatics, University of Belgrade, 11000 Belgrade, Serbia

**Keywords:** prefibrotic primary myelofibrosis (pre-PMF), essential thrombocythemia (ET), differential diagnosis, predictive model

## Abstract

**Simple Summary:**

Essential thrombocythemia and primary myelofibrosis are Philadelphia-chromosome negative myeloproliferative neoplasms with a similar initial phenotypic presentation with thrombocytosis but totally different prognoses and treatment approaches. Patients with primary myelofibrosis compared to essential thrombocythemia had significantly shorter survival as well as higher rates of progression to overt myelofibrosis and leukemic transformation. Around 20% of patients with an early prefibrotic phase of primary myelofibrosis have poor outcomes with a median overall survival of less than 3 years. The aim of our study was to determine significant clinical and laboratory parameters at presentation to differentiate these two entities, according to which we have developed and validated a predictive diagnostic prePMF score. Based on this score, patients with ≥2 points are suspected to have primary myelofibrosis, and hematological work-up should be performed as soon as possible because there is a chance that some of these patients may have rapid disease progression and a shortened life expectancy.

**Abstract:**

Essential thrombocythemia (ET) and prefibrotic primary myelofibrosis (prePMF) initially have a similar phenotypic presentation with thrombocytosis. The aim of our study was to determine significant clinical-laboratory parameters at presentation to differentiate prePMF from ET as well as to develop and validate a predictive diagnostic prePMF model. This retrospective study included 464 patients divided into ET (289 pts) and prePMF (175 pts) groups. The model was built using data from a development cohort (229 pts; 143 ET, 86 prePMF), which was then tested in an internal validation cohort (235 pts; 146 ET, 89 prePMF). The most important prePMF predictors in the multivariate logistic model were age ≥ 60 years (RR = 2.2), splenomegaly (RR = 13.2), and increased lactat-dehidrogenase (RR = 2.8). Risk scores were assigned according to derived relative risk (RR) for age ≥ 60 years (1 point), splenomegaly (2 points), and increased lactat-dehidrogenase (1 point). Positive predictive value (PPV) for pre-PMF diagnosis with a score of ≥points was 69.8%, while for a score of ≥3 it was 88.2%. Diagnostic performance had similar values in the validation cohort. In MPN patients with thrombocytosis at presentation, the application of the new model enables differentiation of pre-PMF from ET, which is clinically relevant considering that these diseases have different prognoses and treatments.

## 1. Introduction

Essential thrombocythemia (ET) and primary myelofibrosis (PMF) are Philadelphia-chromosome negative myeloproliferative neoplasms (Ph-neg MPN) characterized by clonal myeloid proliferation in the bone marrow [1]. The prognoses of ET and PMF are dramatically different: mainly thrombotic complications in ET while symptomatic-aggressive disease with a high risk of transformation to acute myeloid leukaemia in the case of PMF [2]. There are two stages of PMF: early (prefibrotic PMF (prePMF)) and overt advanced, characterized by massive bone marrow (BM) fibrosis, cytopenias, larger splenomegaly, and unfavorable karyotypes with poor prognosis [3].

These two entities, ET and pre-PMF, have a similar initial phenotypic presentation with thrombocytosis [4]. This is why patients sometimes suspected of ET are followed for a long period of time without hematological workup, which leads to the late diagnosis of PMF. Nevertheless, the clinical course of ET and prePMF is totally different. During follow-up of patients with prePMF, in transition to overtPMF, an increase in bone-marrow (BM) fibrosis is seen with the development of extramedullary hematopoiesis in the spleen, accompanied by splenomegaly and reduced platelet counts and hemoglobin values. Patients with prePMF had a significantly higher rate of disease complications, including transformation into acute leukemia and lower survival, than patients with ET. The median overall survival of patients with prePMF is 14.7 years, while that of patients with ET is longer than 20 years [3]. This is why the differentiation between ET and prePMF is so important.

Bone marrow morphology is the gold standard for the diagnosis of MPNs according to the World Health Organization (WHO) criteria from 2016 [1]. The revised WHO classification of myeloid neoplasms from 2016 recognizes prePMF as an individual and distinct entity characterized by well-defined histopathologic features, together with the persistence of at least one of the minor clinical criteria, including leukocytosis ≥11 × $10^{9}$/L, anemia, increased serum lactate dehydrogenase (LDH), and palpable splenomegaly [3]. BM characteristics of prePMF are megakaryocytic proliferation with atypia and reticulin fibrosis ≤ grade 1, accompanied by increased granulocytic proliferation and often decreased erythropoiesis, while in ET, there is isolated megakaryocytic proliferation without atypia and increased granulopoiesis or erythropoiesis or, rarely, BM fibrosis grade 1 [1]. Nevertheless, the lack of fibrosis in the early phases of PMF and clinical onset sometimes characterized by isolated thrombocytosis can lead to prePMF being misdiagnosed as ET [4]. Moreover, some authors have suggested the importance of the subjectivity of histopathological diagnosis, especially in cases of ET and prePMF [1,5,6,7].

Clinical and laboratory characteristics of prePMF patients compared to patients with ET are lower values of hemoglobin, elevated leukocytes and platelets, and higher serum LDH values and frequencies of palpable splenomegaly [8,9,10]. With regard to driver mutations, JAK2V617F and CALR may be present in both ET and prePMF; however, CALR is more associated with megakaryocyte abnormalities and pre-PMF [5].

The aim of this study was to develop and validate a prognostic model utilizing clinical-laboratory characteristics that correlate with the diagnosis of prePMF, which is simple and easily applicable for everyday clinical practice.

## 2. Methods


*Study population*


Here, we present a retrospective analysis of 464 consecutive patients with a diagnosis of ET and pre-PMF, confirmed by histopathology. All patients were consecutively diagnosed and treated at the Clinic of Hematology, University Clinical Center Serbia (UCCS), from January 2015 to December 2018. The study was performed in accordance with the Declaration of Helsinki after approval by the Institutional Ethical Committee.

The final diagnosis was revised according to diagnostic criteria for ET and prePMF established by the WHO classification from 2016 [3]. These criteria include histopathology of bone marrow specimens and clinical (anemia, splenomegaly, increased LDH value and leukocytosis ≥11 × $10^{9}$/L) and molecular data. Only patients with a complete data set, consistent diagnosis of bone marrow morphology, and clinical findings were included in the study.

For the purpose of this study, data from the time of diagnosis were extracted and evaluated from the medical records retrospectively. These data included demographic characteristics (age, gender), the presence of constitutional symptoms (weight loss >10% in 6 months, night sweats, unexplained fever), liver and spleen size by palpation and ultrasound (in cm), blood counts, blasts in the peripheral blood, serum LDH level, grade of BM fibrosis, and molecular-cytogenetic studies. Liver size >15 cm and spleen size >12 cm by ultrasound are values above which the liver and spleen were considered enlarged. Normal LDH values, according to the UCCS central laboratory, are 220–460 U/L and values above 460 U/L are considered increased. Normal values of WBC count are 4–10 × $10^{9}$/L.


*Predictive model development*


The study population was divided based on a split-sample random method into model training (*n* = 229) and internal validation cohorts (*n* = 235). Both cohorts were compared according to the clinical and laboratory variables at time of diagnosis, and the results are shown in Table 1. The differences in proportions between the training and internal validation cohorts were computed using the Wilson approach, which incorporates continuity correction. These differences were then reported alongside their corresponding 95% confidence intervals (CI) [11,12]. The model was constructed with data solely from a training group including 229 patients. The evaluation of variables was conducted using a univariate logistic regression analysis, while the development of the model was achieved using stepwise multivariate logistic regression analysis. The selection of variables was based on their significant relationships with prePMF (*p* < 0.05; determined through univariate analysis) or their known importance, and these variables were included in the variable pool for the stepwise-regression model. The covariates were assessed for collinearity using the variance inflation factor (VIF). The predictive score for the diagnosis of prePMF was calculated using regression coefficients from the final multivariate model. Once a final model was defined, patients with 2 or more points were predicted to have prePMF.


*Internal validation model*


The validation cohort, consisting of 235 patients, was used to evaluate the final model. The definitions and measurements in the validation group were consistent with those of the training group. The evaluation of model discrimination performance was conducted by employing commonly used metrics such as sensitivity, specificity, positive predictive value, and negative predictive value. To conduct a comprehensive evaluation, discrimination was assessed utilizing the C-statistic, which quantifies the area under the receiver operating characteristic curve. Higher values of the C-statistic imply enhanced discrimination. In order to improve the documentation of the development and validation of our risk prediction score system, we employed the TRIPOD checklist. This checklist is a scientifically supported, essential collection of guidelines for documenting prediction modeling studies in the field of biomedical sciences [13]. Statistical analysis was performed using SPSS version 21 statistical software (Chicago, IL, USA).

## 3. Results

The demographics and clinical and laboratory parameters at diagnosis of 464 patients with MPN and thrombocytosis, who were divided into two groups according to the 2016 WHO criteria (289 ET patients and 175 prePMF), are shown in Table 2. The mean patient age of the whole cohort was 58 years (range, 18–86 years), with 64.7% women. The mean liver size was 12.9 cm (IQR 18), while the spleen size was 11 cm (IQR 20). The laboratory mean values were as follows: hemoglobin 138 g/L (IQR 18), leukocytes 9.9 × $10^{9}$/L (IQR 4.1), platelets 957 × $10^{9}$/L (IQR 472), and LDH 472 U/L (IQR 177).

In the whole cohort, patients with prePMF were older, had lower hemoglobin values, higher leukocyte counts and LDH values, and more frequently showed the presence of hepatomegaly and splenomegaly than patients with ET (Table 2). Platelet count and presentation of JAK2V617F mutation did not significantly differ between these two entities.

A predictive model was constructed in the training cohort based on 229 patients and confirmed in the validation internal group of 235 patients. The distribution of patients according to diagnosis in the training cohort was 62.4% ET (143) and 37.6% prePMF (86), similar to the distribution of patients in the validation internal cohort, which consisted of 62.1% ET (146) and 37.9% prePMF (89) patients. In the training cohort, the mean age of patients with prePMF was 62 years (range, 23–86 years), while that of patients with ET was 55 years (range, 18–83 years), the same as in the validation cohort.

A univariate logistic regression model analysis revealed the importance of the following variables for the diagnosis of prePMF: age ≥ 60 years (RR = 2.083; *p* = 0.009), male sex (RR = 1.668; *p* = 0.067), hepatomegaly (RR = 6.84; *p* = 0.004), splenomegaly (RR = 18.807; *p* < 0.001), hemoglobin < 140 g/L (male) or hemoglobin < 120 g/L (female) (RR = 2.828; *p* = 0.002), leucocytosis ≥ 11 × $10^{9}$/L (RR = 1.949; *p* = 0.016), and increased serum LDH value (RR = 6.222; *p* < 0.001) (Table 3).

The following variables were independently associated with the diagnosis of prePMF in multivariable logistic regression analysis: age ≥ 60 years (RR = 2.2; *p* = 0.026), splenomegaly (RR = 13.2; *p* < 0.001), and increased serum LDH value (RR = 2.8; *p* = 0.003) (Table 3). The regression coefficients obtained from the final model were utilized to allocate points for the development of the predictive prePMF score. This score encompassed all the factors that were found to be statistically significant in the multivariate analysis. The prePMF was made by assigning 1 point for significant variables age ≥60 years and increased LDH value and 2 points for splenomegaly, which is variable with a multivariate analysis of derived relative risk (RR) > 5 (Table 4).

A score of ≥2 points had a positive predictive value (PPV) for the diagnosis of prePMF of 69.8%, while a score of ≥3 had a PPV of 88.2% (Table 5). The diagnostic performance measures exhibited consistent values in the validation cohort. The new easily applicable prePMF predictive model has a significant correlation with bone marrow pathohistological findings.

## 4. Discussion

Our study determined the importance of certain clinical and laboratory parameters that independently predict the diagnosis of prePMF. These parameters were used to create a simple model for everyday clinical practice to identify patients with prePMF among patients with thrombocytosis who require detailed and on-time hematological work-up. The diagnostic performance measures exhibited consistent values in the validation cohort.

Separating ET from pre-PMF with similar phenotypic characteristics to thrombocytosis at presentation is of utmost importance, because they have different prognoses and therapeutic approaches [15]. Patients with prePMF compared with ET had significantly worse 10-year survival (76% and 89%, respectively) and 15-year survival (59% and 80%, respectively), rates of leukemic transformation at 10 years (5.8% and 0.7%, respectively) and 15 years (11.7% and 2.1%, respectively), and rates of progression to overt myelofibrosis at 10 years (12.3% and 0.8%, respectively) and 15 years (16.9% and 9.3%) [8]. A recent study of 130 newly diagnosed prePMF patients showed that the median overall survival of these patients was 68 months (42–94 months), with 20% of patients having poor outcomes and a median overall survival of less than 3 years [16]. This would mean that, in patients with persistent thrombocytosis and suspected prePMF, hematological work-up should be performed as soon as possible because there is a chance that some patients may have rapid disease progression and a shortened life expectancy.

How should the differential diagnosis between ET and pre-PMF be made?

On the one hand, the histological diagnosis of bone marrow biopsy should play a central role in the diagnosis of MPN [17]. In the largest multicenter study, which included 1104 patients with a diagnosis of ET who underwent revision of their diagnostic bone marrow biopsy samples, the results showed that 16% of them were reclassified as pre-PMF [6]. In another study, similar results were found, and the re-evaluation of bone marrow biopsy samples according to the 2016 WHO criteria revealed that prePMF accounted for 14% of patients who were previously diagnosed with ET [18,19,20]. This indicates a central role of one or more hematopathologists for the precise diagnosis of the MPN type according to morphological findings in bone marrow [17]. On the other hand, according to Vardiman et al., the histological criteria in the absence of clinical and other analytical data are insufficient to establish the diagnosis of MPN [21]. Despite the improvement of criteria of histopathological diagnosis, there are still some differences mainly attributed to the experience of the pathologists. According to previous studies, the degree of agreement between histopathologists ranged from moderate to good [3,6,8,22,23,24].

The question is whether it is possible to improve the differential diagnosis between ET and pre-PMF using other laboratory data.

In a study by Thiele et al., patients with prePMF were significantly older (median age 66 years) than patients with ET (median age 57 years) [25]. Patients with prePMF had lower hemoglobin values, higher leukocyte count and serum LDH value, and higher frequencies of palpable splenomegaly than patients with ET [25]. Platelet count was not significantly different between these two groups. The results of our study correspond to the mentioned results. Only 8.8% of prePMF patients did not have at least one of the minor clinical criteria required for 2016 WHO diagnosis [10], which include the presence of at least one of the mentioned criteria: leukocytosis ≥11 × $10^{9}$/L, anemia, increased serum lactate dehydrogenase (LDH), and palpable splenomegaly. This indicates that the minor WHO criteria are important in differentiating these two entities because less than 9% of patients without the presence of minor criteria may be initially difficult to separate without pathohistological BM analysis.

In a study by Kamiunten A et al., a higher serum LDH value and a higher frequency of palpable splenomegaly were associated with the diagnosis of prePMF, but other parameters, such as hemoglobin value and leukocyte and platelet count, were similar between the two groups [18]. In our study, the univariate analysis showed that patients with prePMF had lower hemoglobin values, larger diameters of the liver and spleen, higher leukocyte count, and higher serum LDH value, but in the multivariate analysis, only age ≥ 60 years, increased LDH value, and splenomegaly retained significance. Our analysis revealed that the presence of splenomegaly is the most significant factor in differentiating these two entities.

Studies confirmed the prognostic influence of the grade of marrow fibrosis in patients with PMF [26,27]. Assessment of marrow fibrosis has been shown to have predictive value on survival in PMF. A fibrotic grade of BM (MF > 1) has a negative impact on the survival of patients with PMF (*p* < 0.001), while the risk of death was two-fold higher than in those with a prefibrotic grade of BM [28]. Several studies suggested that pathological findings in the BM were correlated with clinical parameters [29]. There is a more than 65% probability of progression from early prefibrotic to advanced overt PMF accompanied by increasing anemia, splenomegaly, and leukoerythroblastosis [29]. On the contrary, clinical parameters such as anemia result from underlying pathological changes but can also be influenced by extrinsic factors and concomitant diseases, and therefore may not present an indirect method by which to evaluate the disease. Leuko-erythroblastosis>1% is extremely rare, seen in the early phase of PMF. Splenomegaly is one of the major clinical manifestations of PMF and is directly linked to splenic extramedullary hematopoiesis (EMH) [30]. EMH is associated with abnormal trafficking patterns of clonal hematopoietic cells due to the dysregulated bone marrow microenvironment leading to progressive splenomegaly.

In the literature, two predictive models for the diagnosis of prePMF have been published. Carobbio et al. developed a step-by-step algorithm that included hemoglobin values, leukocyte count, and serum LDH level to identify prePMF cases that mimicked ET [31]. The best predictive value was found for higher LDH value. Using this algorithm, almost 50% of patients can be correctly allocated to prePMF but using our score it can be predictive in almost 70% of patients.

Another study, in addition to laboratory parameters, added splenomegaly as a clinical finding in a model designed to distinguish ET from prePMF [14]. They created a model using logistic regression based on an exponential analysis of a formula containing hemoglobin value, leukocyte count, serum LDH level, and the presence of splenomegaly. The most important predictor for discriminating these two entities in this study was also serum LDH level, followed by splenomegaly and leukocyte count. In this formula, log2 stands for binary logarithm. The cut-off in the final model was 0.438, and it was set that sensitivity and specificity were as close as possible for the diagnosis of prePMF. We tested this model in our training and validation cohorts, and it showed very high sensitivity (93–96%) but very low specificity (32–34%) (Table 5) and the use of this exponential formula is difficult for everyday clinical practice. Our prePMF model also showed high sensitivity but also very high specificity ≥80% and, moreover, it is easy for daily clinical use.

## 5. Conclusions

In conclusion, primary myelofibrosis is a disease with a variable course, ranging from an indolent, asymptomatic disease to bone marrow failure or leukemic transformation. PrePMF can mimic ET on presentation. Morphologic examination is necessary for distinguishing these entities, but the analysis of bone marrow biopsy in a minor group of patients is impaired by subjectivity in pathological practice and possible degrees of inter- and intra-operator variability even amongst experienced hematopathologists as previously reported. To avoid that, new approaches to digital pathology using machine learning are being developed to improve the assessment and classification of MPN types that may provide significant support for a more accurate histological diagnosis. Our score does not represent a substitute for pathohistological analysis but only contributes to increasing clinicians’ suspicion of prePMF in a patient with a working clinical diagnosis of ET, which enables timely completion of hematological work-ups with bone marrow biopsy. Also, the use of an easily applicable predictive score based on clinical-laboratory parameters represents an additional tool with which to confirm the diagnosis of PMF in addition to pathohistological analysis, especially in cases where bone marrow fibrosis is absent. Considering that this is a single-center study, which is a limiting factor, confirmation through a larger multicenter study would be desirable.

## Figures and Tables

**Table 1 cancers-15-04180-t001:** Characteristics of patients in training and internal validation cohort.

Patient Characteristics	Training Cohort (*n* = 229)	Internal Validation Cohort (*n* = 235)	Difference	95%Cl for Difference
Age (y),	57.5 ± 15.6	58.6 ± 14.14	−1.153	−3.872 to 1.565
mean (sd)				
Age ≥ 60 (y)	118 (51.5)	122 (51.9)	−0.004	−0.095 to 0.088
N (%)				
Male sex	89 (38.9)	75 (31.9)	0.069	−0.018 to 0.157
N (%)				
Liver size (mm),	130.1 ± 13.5	128.1 ± 17	2.029	−0.781 to 4.839
mean (sd)				
Spleen size (mm),	110.5 ± 24	109.4 ± 23	1.034	−3.233 to 5.300
mean (sd)				
Hemoglobin level (g/L),	138 ± 15	138 ± 14.5	0.385	−2.299 to 3.069
mean (sd)				
Leukocyte count ($10^{9}$/L),	10.3 ± 4.6	11 ± 9.4	−0.7344	−2.0899 to 0.6212
mean (sd)				
Platelet count ($10^{9}$/L),	963 ± 391	952 ± 289	10.294	−52.402 to 72.990
mean (sd)				
LDH level,	470 ± 197	474 ± 194	−4.231	−39.898 to 31.437
mean (sd)				
ET	143 (62.4)	146 (62.1)		
N (%)				
PMF	86 (37.6)	89 (37.9)	−0.003	−0.092 to 0.085
N (%)				

**Table 2 cancers-15-04180-t002:** Categorial variables in training cohort.

Patient Characteristics	ET (*n* = 143)	PMF Cohort (*n* = 86)	*p*
Age ≥ 60 (y)	64 (44.8)	54 (62.8)	0.008
N (%)			
Male sex	49 (34.3)	40 (46.5)	0.066
N (%)			
Hepatomegaly (size > 15 cm)	3 (2.1)	11 (12.8)	0.001
N (%)			
Splenomegaly (size > 12 cm)	9 (6.3)	48 (55.8)	<0.001
N (%)			
Splenomegaly (size > 14 cm)	2 (1.4)	33 (38.4)	<0.001
N (%)			
Hemoglobin level (g/L) (Male < 140 g/L, Female < 120 g/L)	19 (13.3)	26 (30.2)	0.002
N (%)			
Leukocytosis (≥11 × $10^{9}$/L)	38 (26.6)	37 (43.0)	0.001
N (%)			
Thrombocytosis (≥$450\times 10^{9}$ /L)	143 (100)	85 (98.8)	0.437
N (%)			
Increased LDH level *	33 (23.1)	56 (65.1)	<0.001
N (%)			

* Lactat dehydrogenase (LDH) (increased LDH is ≥460 U/L according to institutional central laboratory).

**Table 3 cancers-15-04180-t003:** Factors at presentation associated with diagnosis of prePMF (univariate and multivariate logistic regression model).

Risk Factor	Univariate			Multivariate		
	* **p** *	**RR**	**95%CI**	* **p** *	**RR**	**95%CI**
Age ≥ 60 (y)	*p* = 0.009	2.083	1.205–3.602	*p* = 0.026	2.181	1.096–4.339
Male gender	*p* = 0.067	1.668	0.966–2.882			
Hepatomegaly	*p* = 0.004	6.844	1.852–25.292			
Splenomegaly	*p* < 0.001	18.807	8.468–41.767	*p* < 0.001	13.255	5.635–31.178
Hemoglobin (<140 g/L male, <120 g/L female)	*p* = 0.002	2.828	1.451–5.510			
Leukocytosis (≥11 × $10^{9}$/L)	*p* = 0.016	1.949	1.130–3.360			
Increased LDH * value	*p* < 0.001	6.222	3.450–11.224	*p* = 0.003	2.858	1.430–5.713

* LDH—serum lactat dehydrogenase value.

**Table 4 cancers-15-04180-t004:** Predictive model for diagnosis of prePMF.

Parameter	Assigned Score
Age ≥ 60 (y)	1
Splenomegaly	2
Increased LDH value	1

**Table 5 cancers-15-04180-t005:** Accuracy and validation of prediction score for pre-PMF.

	Training Cohort				Internal Validation Cohort			
	**Sn**	**Sp**	**PPV**	**NPV**	**Sn**	**Sp**	**PPV**	**NPV**
Score +2	0.698	0.818	0.698	0.818	0.685	0.747	0.622	0.796
Score +3	0.523	0.958	0.882	0.77	0.404	0.952	0.837	0.724
Regression model * >0.438	0.965	0.322	0.461	0.939	0.933	0.342	0.464	0.893

* [14].

## Data Availability

All data regarding this research are available upon reasonable request to the corresponding author.

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
