# Peer review of "Easily Applicable Predictive Score for Differential Diagnosis of Prefibrotic Primary Myelofibrosis from Essential Thrombocythemia"

_cancers, 2023, doi:10.3390/cancers15164180_

Round 1

Reviewer 1 Report

The paper presented here is of sound scientific merit. Indeed, the predictive score is easy to use, and could prove to be helpful in everyday practice, at least in the early phases of diagnostic work up.

The sentence starting in the row 166  should be revised, it is not clear what it is suggested here: patient should not be followed for a long time, or did the authors mean that such patients should be followed more frequently?

Also, I urge the authors to reconsider conclusion, namely the part where it is stated that the bone marrow biopsy is sometimes insufficient (and that is further elaborated) especially after in Introduction it is stated that bone marrow morphology is the gold standard in differentiating these two diagnoses. 

The predictive score can have its place but it is far from replacing morphology.

The sentence in line 166 should be revised since it is not easy to understand what its message is.

Author Response

Comment 1: The paper presented here is of sound scientific merit. Indeed, the predictive score is easy to use, and could prove to be helpful in everyday practice, at least in the early phases of diagnostic work up.

The sentence starting in the row 166 should be revised, it is not clear what it is suggested here: patient should not be followed for a long time, or did the authors mean that such patients should be followed more frequently?

Response 1. Original sentence (row 166): “This would mean that patients with persistent thrombocytosis and suspected prePMF should not be followed for a long period because there is a chance that some patients may have rapid disease progression and a shortened life expectancy”

Thank you for your comment. We corrected this sentence.

Corrected sentence: “This would mean in patients with persistent thrombocytosis and suspected prePMF, hematological work-up should be performed as soon as possible because there is a chance that some patients may have rapid disease progression and a shortened life expectancy”.

Comment 2: Also, I urge the authors to reconsider conclusion, namely the part where it is stated that the bone marrow biopsy is sometimes insufficient (and that is further elaborated) especially after in Introduction it is stated that bone marrow morphology is the gold standard in differentiating these two diagnoses. 

The predictive score can have its place but it is far from replacing morphology.

Original conclusion in manuscript: “In conclusion, primary myelofibrosis is a disease with a variable course, ranging from an indolent, asymptomatic disease to bone marrow failure or leukemic transformation. PrePMF can mimic ET on presentation, and morphologic examination is necessary for distinguishing these entities, but bone marrow biopsy is sometimes insufficient because it is impaired by subjectivity in pathological practice and is of questionable clinical relevance due to high degrees of inter- and intra-operator variability even amongst experienced hematopathologists earlier reported, at least when considering individual patient. The use of easy applicable predictive score based on clinical-laboratory parameters can provide clinicians to increase suspicion of prePMF in a patient with a working clinical diagnosis of ET, which enables timely haematological work-up. New approaches of digital pathology using machine learning are being developed to improve the assessment and classification of MPNs types that may in the near future provide a significant support for a more accurate histological diagnosis”.

Response 2: Thank you for your comment and we have corrected the conclusion accordingly. In the introduction, it is indicated that the pathohistological analysis represents the gold standard for establishing the diagnosis, but also, as stated in the previous manuscripts, it is based also on subjectivity of hematopathologists. Our score is not a substitute for pathohistological analysis, but helps in identifying patients who require complete hematological work-up in the direction of PMF without delay, which also includes bone marrow analysis.

Revised conclusion:” In conclusion, primary myelofibrosis is a disease with a variable course, ranging from an indolent, asymptomatic disease to bone marrow failure or leukemic transformation. PrePMF can mimic ET on presentation. Morphologic examination is necessary for distinguishing these entities, but analysis of bone marrow biopsy in minor group of patients is impaired by subjectivity in pathological practice and possible degrees of inter- and intra-operator variability even amongst experienced hematopathologists as previously reported. To avoid that, new approaches of digital pathology using machine learning are being developed to improve the assessment and classification of MPNs types that may provide a significant support for a more accurate histological diagnosis. Our score does not represent a substitute for pathohistological analysis but only contributes clinicians to increase suspicion of prePMF in a patient with a working clinical diagnosis of ET, which enables timely complete haematological work-up with bone marrow biopsy. Also, the use of an easily applicable predictive score based on clinical-laboratory parameters represents an additional tool to confirm the diagnosis of PMF in addition to pathohistological analysis, especially in cases where bone marrow fibrosis is absent.Considering that this is a single-center study, which is a limiting factor, confirmation through a larger multicenter study would be desirable”.

Reviewer 2 Report

In their paper, Lekovic et al. aimed to construct a prognostic score which could help to improve the differentiation of ET vs pre-MF using easily available clinical and laboratory variables. The paper tackles an important clinical issue, is executed well, clearly written and organized.  My comments are as follows: 

1) My main concern is the novelty of the presented results. For example, we know that elevated LDH and palpable spleen may be indicative of possible pre-MF/MF - these variable are already included in the WHO criteria for pre-MF as "minor criteria". We also know that bone marrow biopsy is needed for all ET patients. Also, around 20% of the patients seem not to be correctly classified with the presented score.  The authors should, therefore, discuss and show to the readers that their score is also prognostically relevant and important. For example, do they have data on survival of these patients? Does this score  discriminate survival of these patients? Do age, LDH and spleen differ in their prognostic impact?  Could some patients with score <2 perhaps be spared of diagnostic bone marrow biopsy even though they have pre-MF? This is especially important for those older than 60 years of age, who are most probably not allotransplant candidates... This "twist" is somehow missing.

2) Why hazard ratios (HR) are reported when logistic regression is used ? And in the table risk ratio-RR is shown?

3) Were data collected at disease diagnosis? This is not clear from the Methods section. 

4) How was bone marrow evaluated? How many pathologists were included in the bone marrow reviews? Please elaborate more. 

4) Some typos and polishing is needed, ie, score >2 missing in the Abstract, "uncenter", "respect" should be "with respect", and so on. Please read carefully.

English is good. 

Author Response

Review 2: In their paper, Lekovic et al. aimed to construct a prognostic score which could help to improve the differentiation of ET vs pre-MF using easily available clinical and laboratory variables. The paper tackles an important clinical issue, is executed well, clearly written and organized.  My comments are as follows:

  • My main concern is the novelty of the presented results. For example, we know that elevated LDH and palpable spleen may be indicative of possible pre-MF/MF - these variable are already included in the WHO criteria for pre-MF as "minor criteria". We also know that bone marrow biopsy is needed for all ET patients. Also, around 20% of the patients seem not to be correctly classified with the presented score. The authors should, therefore, discuss and show to the readers that their score is also prognostically relevant and important. For example, do they have data on survival of these patients? Does this score discriminate survival of these patients? Do age, LDH and spleen differ in their prognostic impact?  Could some patients with score <2 perhaps be spared of diagnostic bone marrow biopsy even though they have pre-MF? This is especially important for those older than 60 years of age, who are most probably not allotransplant candidates... This "twist" is somehow missing.

Response 1. The sensitivity and specificity of a prognostic score of 70% or more is considered extremely significant. Our score has sensitivity and specificity over 80%. This score is not a substitute for pathohistological analysis, but helps in identifying patients who require complete hematological work-up in the direction of PMF without delay, which also includes bone marrow analysis. Also, the use of easily applicable predictive score based on clinical-laboratory parameters represent additional tool to confirm the diagnosis of PMF in addition to pathohistological analysis especially in cases where bone marrow fibrosis is absent. We added another whole paragraph and corrected the conclusion. All changes in the manuscript are marked in red.

  • Why hazard ratios (HR) are reported when logistic regression is used? And in the table risk ratio-RR is shown?

Thank you for your comment. Everything is adjusted according to logistic regression as it should be.

  • Were data collected at disease diagnosis? This is not clear from the Methods section.

Thank you for your comment. Added in two places including the methodology and highlighted in red.

  • How was bone marrow evaluated? How many pathologists were included in the bone marrow reviews?

In several places in the paper, there are comments regarding bone marrow biopsy in MPN. In our clinic, we get the final report from our hematopathology department and they coordinate the analysis of the bone marrow biopsies between them. Our two hematopathologists are highly specialized in MPNs, one of them was in Barbara Bui's lab and the other was in Jüergen Thiele's and they consult with each other.

  • Some typos and polishing is needed, ie, score >2 missing in the Abstract, "uncenter", "respect" should be "with respect", and so on. Please read carefully

Thank you for your comment. We have corrected written errors that we have noticed. The paper must be prepared in LaTeX program with specific software, which additionally complicates the situation for the authors.